# Combinations of Host- and Virus-Targeting Antiviral Drugs Confer Synergistic Suppression of SARS-CoV-2

Jessica Wagoner,[a] Shawn Herring,[a] Tien-Ying Hsiang,[b] Aleksandr Ianevski,[c] Scott B. Biering,[d] Shuang Xu,[e] Markus Hoffmann,[f,g] Stefan Pöhlmann,[f,g] Michael Gale, Jr.,[b] Tero Aittokallio,[c,h] Joshua T. Schiffer,[e,i,j] Judith M. White,[k,l] Stephen J. Polyak[a,m,n]

[a]Virology Division, Department of Laboratory Medicine and Pathology, University of Washington, Seattle, Washington, USA
[b]Department of Immunology, University of Washington, Seattle, Washington, USA
[c]Institute for Molecular Medicine Finland (FIMM), HiLIFE, University of Helsinki, Helsinki, Finland
[d]Division of Infectious Diseases and Vaccinology, School of Public Health, University of California—Berkeley, Berkeley, California, USA
[e]Vaccine and Infectious Diseases Division, Fred Hutchinson Cancer Research Center, Seattle, Washington, USA
[f]Infection Biology Unit, German Primate Center, Leibniz Institute for Primate Research, Göttingen, Germany
[g]Faculty of Biology and Psychology, University of Göttingen, Göttingen, Germany
[h]Oslo Centre for Biostatistics and Epidemiology, University of Oslo and Oslo University Hospital, Oslo, Norway
[i]Division of Allergy and Infectious Disease, University of Washington, Seattle, Washington, USA
[j]Clinical Research Division, Fred Hutchinson Cancer Research Center, Seattle, Washington, USA
[k]Department of Cell Biology, University of Virginia, Charlottesville, Virginia, USA
[l]Department of Microbiology, University of Virginia, Charlottesville, Virginia, USA
[m]Department of Global Health, University of Washington, Seattle, Washington, USA
[n]Department of Microbiology, University of Washington, Seattle, Washington, USA

**ABSTRACT** Three directly acting antivirals (DAAs) demonstrated substantial reduction in COVID-19 hospitalizations and deaths in clinical trials. However, these agents did not completely prevent severe illness and are associated with cases of rebound illness and viral shedding. Combination regimens can enhance antiviral potency, reduce the emergence of drug-resistant variants, and lower the dose of each component in the combination. Concurrently targeting virus entry and virus replication offers opportunities to discover synergistic drug combinations. While combination antiviral drug treatments are standard for chronic RNA virus infections, no antiviral combination therapy has been approved for SARS-CoV-2. Here, we demonstrate that combining host-targeting antivirals (HTAs) that target TMPRSS2 and hence SARS-CoV-2 entry, with the DAA molnupiravir, which targets SARS-CoV-2 replication, synergistically suppresses SARS-CoV-2 infection in Calu-3 lung epithelial cells. Strong synergy was observed when molnupiravir, an oral drug, was combined with three TMPRSS2 (HTA) oral or inhaled inhibitors: camostat, avoralstat, or nafamostat. The combination of camostat plus molnupiravir was also effective against the beta and delta variants of concern. The pyrimidine biosynthesis inhibitor brequinar combined with molnupiravir also conferred robust synergistic inhibition. These HTA+DAA combinations had similar potency to the synergistic all-DAA combination of molnupiravir plus nirmatrelvir, the protease inhibitor found in paxlovid. Pharmacodynamic modeling allowed estimates of antiviral potency at all possible concentrations of each agent within plausible therapeutic ranges, suggesting possible *in vivo* efficacy. The triple combination of camostat, brequinar, and molnupiravir further increased antiviral potency. These findings support the development of HTA+DAA combinations for pandemic response and preparedness.

**IMPORTANCE** Imagine a future viral pandemic where if you test positive for the new virus, you can quickly take some medicines at home for a few days so that you do not get too sick. To date, only single drugs have been approved for outpatient use against SARS-CoV-2, and we are learning that these have some limitations and may succumb to drug resistance. Here, we show that combinations of two oral drugs are

Address correspondence to Stephen J. Polyak, polyak@uw.edu.

The authors declare no conflict of interest.

better than the single ones in blocking SARS-CoV-2, and we use mathematical modeling to show that these drug combinations are likely to work in people. We also show that a combination of three oral drugs works even better at eradicating the virus. Our findings therefore bode well for the development of oral drug cocktails for at home use at the first sign of an infection by a coronavirus or other emerging viral pathogens.

**KEYWORDS** SARS-CoV-2, coronavirus, host-targeting antiviral, HTA, directly acting antiviral, DAA, combination, synergy, molnupiravir, camostat, paxlovid, Calu-3, antiviral, drug combinations, variant(s) of concern, Ebola

Despite the approval of one intravenous (remdesivir) and two oral (molnupiravir and paxlovid) anti-SARS-CoV-2 drugs, the armamentarium against SARS-CoV-2 and related coronaviruses remains thin. Monoclonal antibodies have proven effective as prophylaxis and therapy (1, 2) but must be given intravenously or subcutaneously in a clinical setting. Initially effective products have rapidly become irrelevant in the face of newly emerging viral variants (3). While highly effective at preventing severe disease if given early during infection in outpatient settings (4), remdesivir has been mostly given during hospitalization (5, 6). Its use during early disease is limited due to its intravenous formulation. Molnupiravir, an oral drug, is 30% effective at preventing hospitalization when given early during infection (7). Paxlovid, another oral drug, is highly efficacious (8), but side effects, drug interactions (9), and rebound of symptoms and viral shedding upon treatment cessation (10, 11) can limit therapy success. The recent descriptions of paxlovid resistance *in vitro* (12–16) suggests there may be at some point no highly effective oral agent to prevent hospitalization in high-risk infected people. Thus, there remains a need to develop SARS-CoV-2 prophylactic and therapeutic regimens that are suitable for self/ home administration, via inhaled or oral routes, particularly for infected people who are newly diagnosed and early in their COVID-19 course (17). Ideally, these treatments should be relatively inexpensive so they can be widely used by infected people and their contacts around the globe.

It is well established that the most effective drug-based therapies for chronic, persistent RNA viruses include a combination of two or three drugs (18, 19). Drug combinations comprise the standard of care for chronic RNA virus infections such as HIV and HCV infections and demonstrate promise for treating acute RNA virus infections, including infections with filoviruses (20–22), arenaviruses (23), influenza viruses (24–27), and, most recently, SARS-CoV-2 (28–31; for a review, see reference 32). A key feature of drug combinations is that they may confer bioactivities beyond additivity, such as multiplicative or synergistic effects, by targeting different steps of the viral life cycle. (Paxlovid is a combination treatment consisting of nirmatrelvir [PF-07321332], which inhibits the SARS-CoV-2 3C-like protease, and ritonavir, which slows the metabolism of nirmatrelvir [33]. However, paxlovid does not leverage dual drug target sites.) Synergistic antiviral activity may permit dose reductions of each drug in the combination, thereby reducing the potential for clinical side effects while allowing increased clinical efficacy. Moreover, synergistic drug combinations can bring *in vitro* drug levels that inhibit virus by 50% ($IC_{50}$) into *in vivo* pharmacokinetic (PK) ranges. For chronic RNA virus infections such as HIV and HCV infections, drug combinations critically limit emergence of drug resistant viral mutants (18, 19).

Combination therapies may be of particular use for immunocompromised patients with SARS-CoV-2 infection. More potent treatments are likely needed for patients who may not be able to exert adequate immune pressure on the virus (34–36). Moreover, SARS-CoV-2 evolution in immunosuppressed patients has become a major public health concern since variants of concern (VOCs) may arise during SARS-CoV-2 infections that persist at high viral load for months (37). Novel VOC continue to extend the pandemic by undermining vaccine efforts. There is also a greater window for the emergence of drug-resistant mutants in this setting.

Oral drug combination regimens are often preferred to allow outpatient treatment early during disease as prompt treatment after symptom onset is associated with better outcomes for SARS-CoV-2 virus (1, 2, 4, 7), influenza virus (38), Ebola virus (39), HIV (40), and zoster virus (41) infections. Here, we demonstrate the synergistic potential of combining host-targeting antivirals (HTAs) that either target SARS-CoV-2 entry (camostat, nafamostat, and avoralstat) or an HTA (brequinar) that targets SARS-CoV-2 replication with a DAA replication inhibitor (molnupiravir) to block SARS-CoV-2 infection in Calu-3 lung epithelial cells. All four agents (three oral and one inhaled) synergized with the oral agent molnupiravir, with the pairs camostat plus molnupiravir and brequinar plus molnupiravir demonstrating the strongest synergy, which was similar to the synergy observed with the all-DAA combination of molnupiravir plus nirmatrelvir. When the three component drugs, brequinar, camostat, and molnupiravir were combined, even greater antiviral efficacy and potency was observed. Pharmacodynamic (PD) modeling of the *in vitro* data suggest drug concentrations at which high potency can be achieved, which may fall within observed plasma values of these agents, hence suggesting possible *in vivo* effects.

## RESULTS

**Choice of drugs for combination testing versus SARS-CoV-2.** We aim to develop a combination of oral or inhaled drugs with potent activity against SARS-CoV-2 in lungs. Towards this end, we explored combined targeting of SARS-CoV-2 entry and replication, as blocking different phases of the SARS-CoV-2 life cycle offers the possibility of synergistic antiviral efficacy (32). Our initial focus drugs targeted TMPRSS2, a host cell serine protease critical for SARS-CoV-2 entry into lung cells (42), and molnupiravir, a potent oral inhibitor of the SARS-CoV-2 RNA-dependent RNA polymerase (43). The TMPRSS2 inhibitors included two approved drugs, camostat and nafamostat, as well as the preclinical drug avoralstat (42, 44, 45). Camostat and avoralstat are both oral drugs, while nafamostat is administered intravenously and is being tested via inhalation for COVID-19. All three drugs have shown efficacy in small animal models of SARS-CoV-2 infection via oral or inhaled routes (45, 46). Both oral camostat and intravenous nafamostat have been studied in small SARS-CoV-2 treatment trials in hospitalized patients, but they have shown no or only small benefits as solo agents (47–49). We considered three additional oral drugs proposed as entry inhibitors: apilimod, arbidol, and imatinib (50–53). However, based on literature values and our studies with VSV pseudovirions expressing the SARS-CoV-2 spike (S) protein, these oral drugs had lower bioavailability ($C_{max}$) compared to their concentration for 50% inhibition ($IC_{50}$) in lung cells than either camostat or avoralstat (see Table S2) and so were not pursued here. In addition to molnupiravir, we added brequinar to our set of drugs for combination testing. Brequinar is a pyrimidine biosynthesis inhibitor that one of us previously showed to synergize with the HCV polymerase inhibitor sofosbuvir to thwart HCV infection (29, 54), and it was also recently shown to synergize with molnupiravir against SARS-CoV-2 (31). As for other cell-based tests of molnupiravir, we employed its active form, EIDD-1931, for our studies.

We first established antiviral dose responses for the compounds as single agents. EIDD-1931, camostat, nafamostat, avoralstat, and brequinar yielded $IC_{50}$s against SARS-CoV-2 of 240 nM, 143 nM, 71.3 nM, 2.6 $\mu$M, and 50 $\mu$M, respectively, in Calu-3 cells (Fig. 1) with minimal toxicity on noninfected cells over the dose ranges studied. To study the cell line specificity of the activities, we also evaluated molnupiravir, camostat, and avoralstat against SARS-CoV-2 infection of a 293T cell line that overexpresses ACE2 and TMPRSS (55). In these 293TAT cells, the $IC_{50}$s against SARS-CoV-2 were 150 nM, 1.2 $\mu$M, and 1.4 $\mu$M for EIDD-1931, camostat, and avoralstat, respectively, similar to their potencies in Calu-3 cells with the exception of camostat, which was more active in Calu-3 compared to 293TAT cells (see Fig. S1 in the supplemental material).

**Combination testing identifies drug pairs with synergistic activity against SARS-CoV-2 in Calu cells.** Checkerboard drug combination assays were then performed with the three TMPRSS2 inhibitors and brequinar, each in combination with

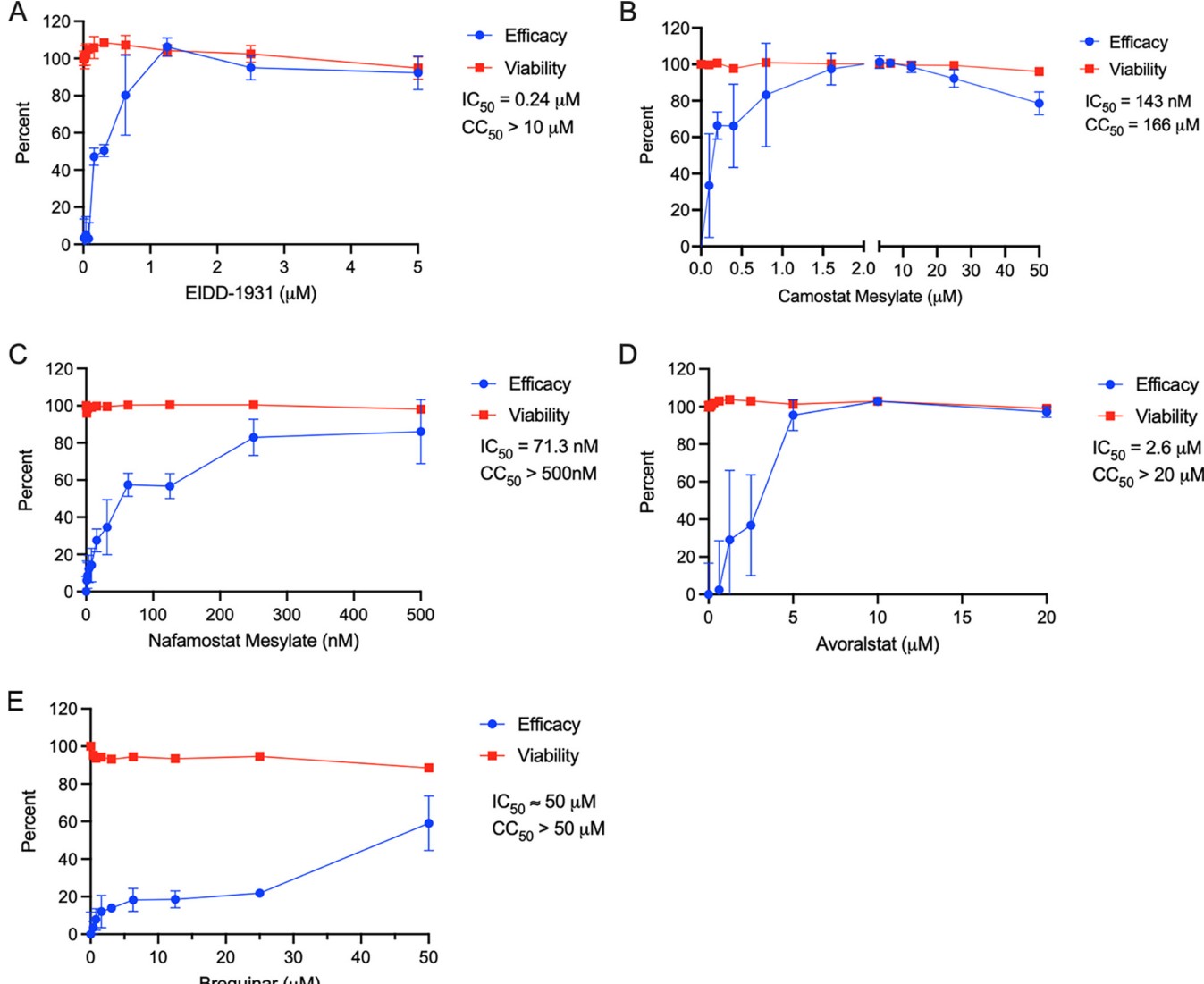

**FIG 1** Inhibition of SARS-CoV-2 infection by DAA and HTA agents. Calu-3 cells were treated with the indicated concentrations of drugs for 2 h prior to infection with SARS-CoV-2 WA1 at an MOI of 0.1. Parallel plates contained cells treated only with drugs to monitor the toxic effects in noninfected cells (viability trace). At 96 h postinfection (72 h for camostat), the cell viability was assessed using a CellTiter-Glo assay, and the antiviral efficacy and viability, expressed as percentages relative to the DMSO solvent control, were calculated as described in Materials and Methods. Data points reflect averages and standard deviations of triplicate samples per condition, and $IC_{50}$ and $CC_{50}$ values were generated by nonlinear regression using Prism. The data represent an independent experiment for each drug. Note that for brequinar, we observed variable $IC_{50}$s (22.4 $\mu$M $\pm$ 24, $n = 3$).

molnupiravir, first in Calu-3 cells. Figure 2 presents representative results as analyzed by SynergyFinder 3.0 (56). All drug combinations conferred dose-dependent synergistic suppression of virus infection. For the combinations of camostat plus molnupiravir, nafamostat plus molnupiravir, and brequinar plus molnupiravir, synergistic suppression of virus infection occurred at 2- to 3-fold lower drug concentrations for the combination compared to single drugs. The observed synergies boost drug effectiveness beyond what would be projected by Bliss independence. For example, for the combination molnupiravir plus camostat, each drug alone maximally provided 81 and 92% inhibitions of infection, respectively. However, the combination provided 99 to 100% inhibition in multiple dose combinations (Fig. 2A). When combined with molnupiravir, camostat and avoralstat showed similar synergistic suppression of SARS-CoV-2 infection of 293TAT cells (Table 1; see also Fig. S3). Thus, the combination effects of the TMPRSS2 inhibitors with molnupiravir extend to multiple human cell lines where virus entry occurs by TMPRSS2-mediated fusion. Table 1 summarizes the data from all drug

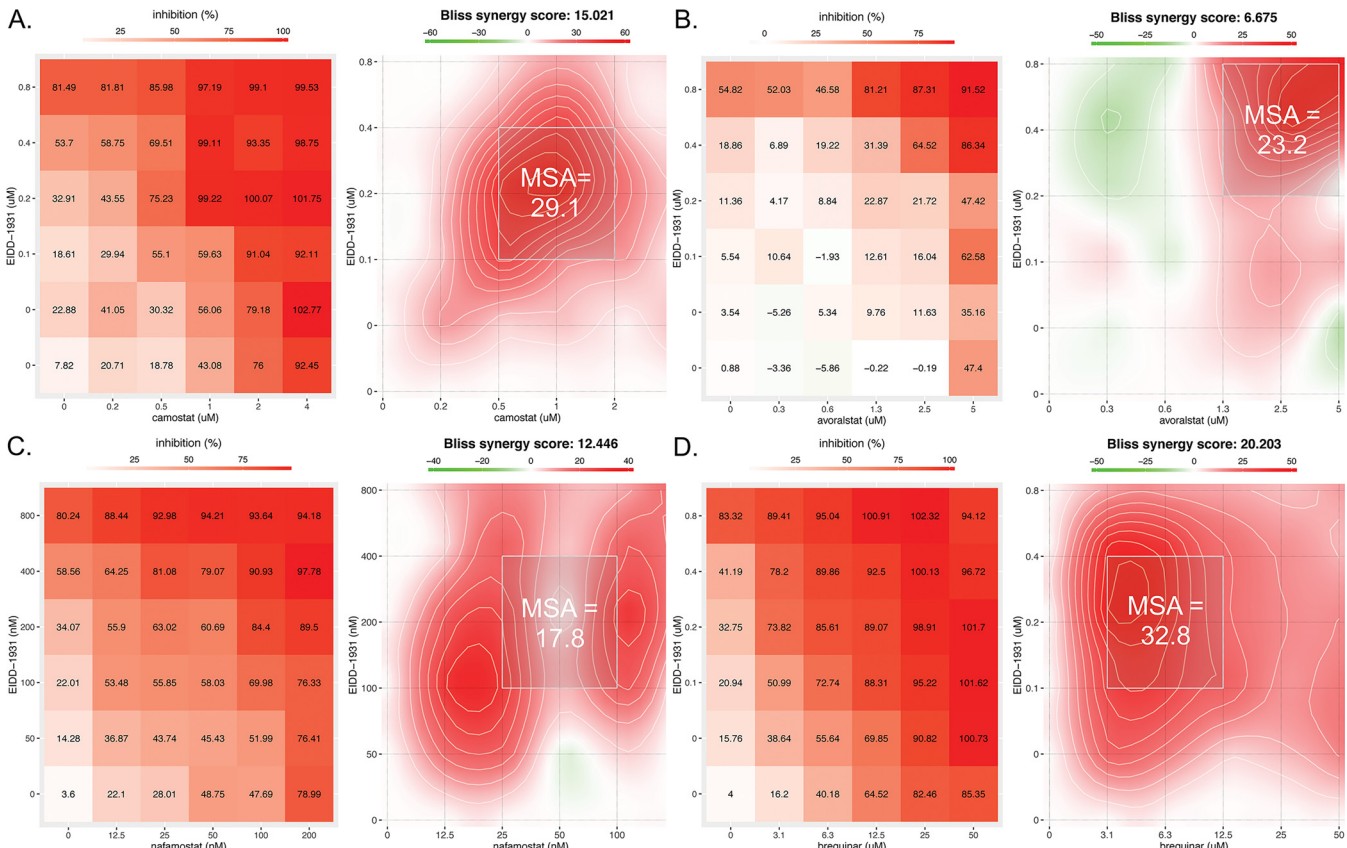

**FIG 2** TMPRSS2 inhibitors and brequinar synergize with molnupiravir to suppress SARS-CoV-2 infection in Calu-3 cells. Calu-3 lung cells were treated with the indicated concentrations of camostat, avoralstat, nafamostat, brequinar, and molnupiravir (EIDD-1931) 2 h prior to infection with SARS-CoV-2 WA1 (MOI of 0.1). After 96 h, the cell viability was measured by a CellTiter-Glo assay (Promega), and the antiviral efficacy was calculated as described in Materials and Methods. For each panel (A through D), the left plot shows the percent inhibition of infection, while the right plot depicts a two-dimensional topograph that highlights the areas of synergy across the full dose response matrix, including the MSA, which is designated by a light gray box. Avoralstat induces maximal synergy at high concentrations of both drugs, whereas the other three compounds induce synergy at lower concentrations of both drugs.

pair experiments. None of the drug combinations in the checkerboard were toxic to non-infected cells, as assessed using SynToxProfiler (57) (see Fig. S2).

Consistent with a recent report (31), combining brequinar with molnupiravir conferred robust, synergistic suppression of SARS-CoV-2 infection in Calu-3 cells (Fig. 2D and Table 1), with average synergy and MSA scores of 22.6 and 33.4, respectively. Synergy occurred at low to moderate drug concentrations, leading to high potency in these regions of the matrix. In these experiments, molnupiravir and brequinar maxi-

**TABLE 1** Summary of combination data against SARS-CoV-2[a]

| Molnupiravir + | Cells | Overall Bliss synergy | MSA | N |
|---|---|---|---|---|
| Camostat | Calu-3 | 13.2 | 26.0 | 2 |
| Avoralstat | Calu-3 | 5.1 | 16.8 | 2 |
| Nafamostat | Calu-3 | 8.7 | 15.0 | 2 |
| Brequinar | Calu-3 | 22.6 | 33.4 | 2 |
| Camostat | 293TAT | 15.6 | 21.3 | 2 |
| Avoralstat | 293TAT | 10.7 | 15.6 | 2 |
| Nirmatrelvir | Calu-3 | 13.0 | 23.8 | 3 |

[a]The overall Bliss synergy represents the average score for the entire 6 × 6 drug combination matrix, while maximum synergistic area (MSA) represents the score for a 3 × 3 submatrix. "N" refers to the number of separate experiments (biological replicates). For Calu-3 cells, all conditions in the checkerboard in each experiment were performed in triplicate, while for HEK293T cells overexpressing TMRPSS2 and ACE2 (293TAT), all conditions in the checkerboard were performed in duplicate in each experiment. Synergy scores and MSAs represent the averages of the replicated experiments. For all drug combinations, WA1 was the challenge virus, except for nirmatrelvir plus molnupiravir, where the delta VOC was the challenge virus.

mally provided ~80% inhibition of infection as solo agents, whereas when applied together they provided 95 to 100% inhibition at multiple combinations of the drugs (Fig. 2D).

To study whether the combinations show activity also against other variants, we next tested the combination of camostat plus molnupiravir against infection of Calu-3 cells by the beta and delta VOCs. This HTA+DAA combination conferred synergistic suppression of both VOCs (Fig. 3A and B). These combination effects were similar to the all-DAA combination of molnupiravir plus nirmatrelvir (Fig. 3C and Table 1).

**Predictive modeling of drug combination potency against SARS-CoV-2 in Calu-3 cells.** To recapitulate observed experimental data and then make projections about combinatorial drug potency at concentrations that were not measured experimentally, we applied an *in vitro* PD model previously validated against antiviral drug pairs (58). The model was tested against all four anti-SARS-CoV-2 DAA+HTA drug pairs in the present study: molnupiravir plus camostat, molnupiravir plus brequinar, molnupiravir plus avoralstat, and molnupiravir plus nafamostat. The model was assessed for fit to the empirically derived drug matrices in which percent inhibition of infection was assessed at various drug concentrations (left plots in Fig. 2A to D).

The mathematical model demonstrated high predictive power by closely projecting the *in vitro* efficacy of drug combinations across the dose response matrices, with an $R^2$ value of $\geq 0.945$ (Table 2). Exponent "*a*" is an approximation for overall synergy across dose response matrices. The drug pair molnupiravir plus brequinar has the highest exponent, indicating most synergistic interaction between the two drugs on average compared to other drug pairs, which is consistent with its highest MSA score (Fig. 2) among the DAA+HTA pairs tested. The model suggests that brequinar leads to synergy by lowering $IC_{50}$ of molnupiravir more than camostat, avoralstat, or nafamostat (Table 2); e.g., the $IC_{50}$ for molnupiravir in the brequinar combination is 0.09877 $\mu$M, whereas it is 0.1383, 0.1883, and 0.2866 $\mu$M, respectively, in the camostat, avoralstat, and nafamostat combinations.

We next explored the mechanisms explaining the model fit to the data by focusing on individual drug combinations. For molnupiravir plus camostat, each drug lacked complete potency when modeled alone even at high concentrations (79.69% inhibition for molnupiravir [Fig. 4B] and 94.02% inhibition for camostat [Fig. 4C]). However, low concentrations of molnupiravir significantly boosted potency to >94% in the presence of high concentrations of camostat (4 $\mu$M, Fig. 4D). This enhanced potency is largely attributable to Bliss independence (i.e., multiplicative effects). Synergy boosts combinatorial effectiveness beyond predicted multiplicative effects at lower concentrations of camostat, particularly at 0.25 $\mu$M (Fig. 4D). Moreover, efficacies approaching 100% are observed across a large swath of the dose matrix (Fig. 4D, yellow).

For molnupiravir plus brequinar, each drug lacked complete potency when modeled alone even at high concentrations (63.93% inhibition for molnupiravir [Fig. 5B] and 63.72% inhibition for brequinar [Fig. 5C]). However, low concentrations of molnupiravir significantly boosted potency given high concentrations of brequinar (50 $\mu$M) (Fig. 5A). This enhanced potency is largely attributable to synergy beyond predicted Bliss independence (multiplicative effects) particularly at low concentrations of molnupiravir (Fig. 5D).

Using model output, we generated heatmaps demonstrating the predicted antiviral potency at all possible combinations of drug concentrations, including those not specifically measured experimentally (Fig. 4D and Fig. 5D; see also Fig. S4D and S5D). The model output can therefore be used to project minute to minute *in vivo* combinatorial antiviral potency of two drugs that have different PK properties of expansion and decay in a person. The data suggest that the combinations may have antiviral potency *in vivo* if observed concentrations can be reached and maintained at the site of infection.

**Activity of a triple drug combination against SARS-CoV-2 in Calu-3 cells.** Since camostat, brequinar, and molnupiravir inhibit distinct cellular and viral targets and both HTA drugs (camostat and brequinar) synergize with the DAA molnupiravir in pairs, we compared the antiviral efficacy of the triple combination versus the three two-drug combinations

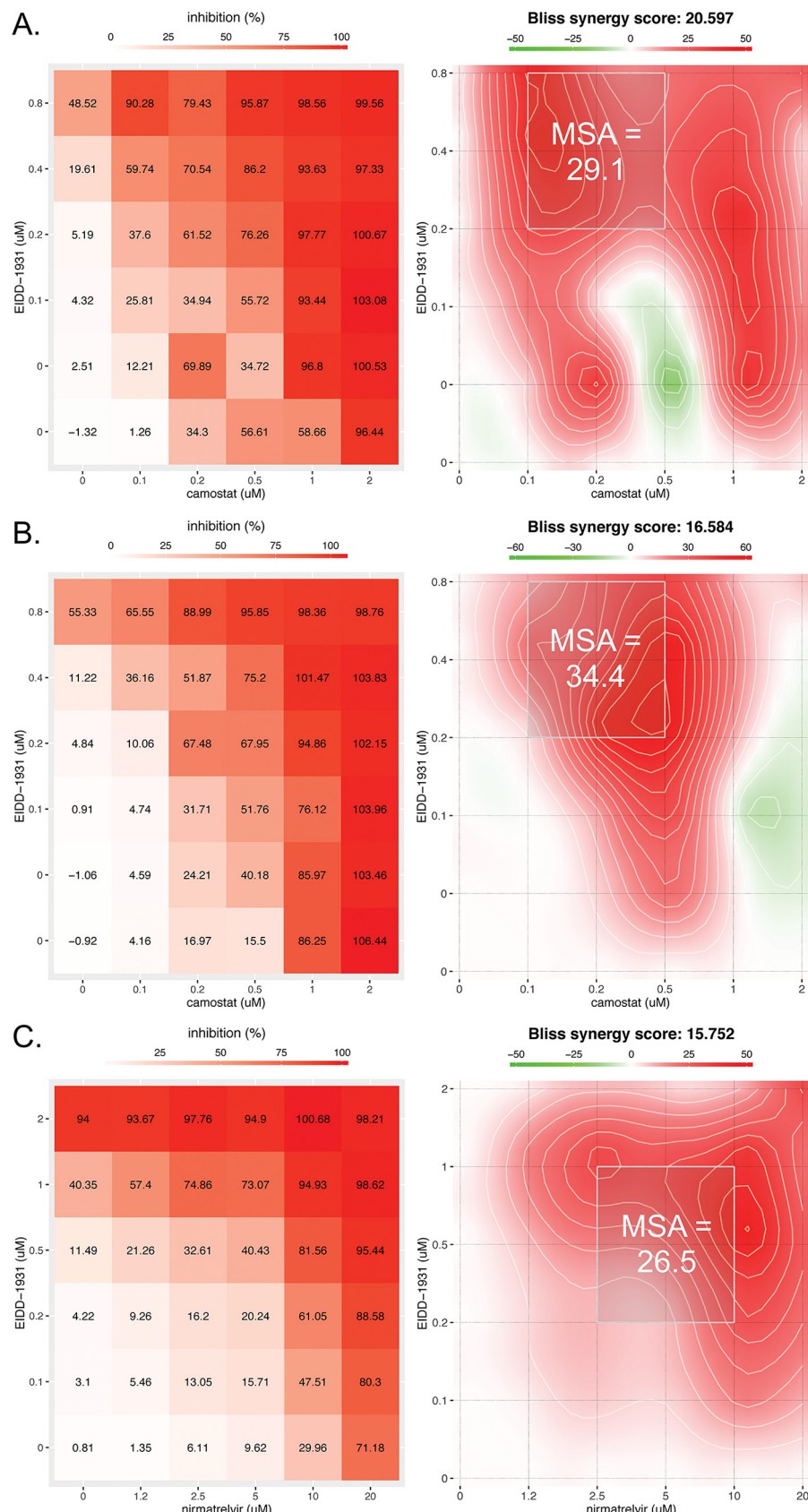

**FIG 3** Camostat and nirmatrelvir synergize with molnupiravir to suppress infection of Calu-3 cells by SARS-CoV-2 VOCs. Calu-3 lung cells were treated with the indicated concentrations of camostat and

**TABLE 2** PD model parameters

| Parameter[a] | Estimated value (95% CI) | | | |
|---|---|---|---|---|
| | Molnupiravir + camostat | Molnupiravir + brequinar | Molnupiravir + avoralstat | Molnupiravir + nafamostat |
| $a$ | 1.787 (0.7468–2.827) | 3.85 (2.923–4.778) | 3.314 (0.9883–5.639) | 1.061 (0.6793–1.443) |
| $h_{EIDD}$ | 1.139 (0.8407–1.437) | 1.001 (0.8819–1.12) | 1.181 (0.8145–1.548) | 1.026 (0.8033–1.249) |
| $h_2$ | 1.662 (1.257–2.066) | 0.7289 (0.6431–0.8146) | 1.748 (1.307–2.188) | 0.9372 (0.7622–1.112) |
| $IC_{50}$ ($\mu$M) | | | | |
| $\quad IC_{50,EIDD}$ | 0.1383 (0.05129–0.2253) | 0.09877 (0.07293–0.1246) | 0.1883 (0.04864–0.328) | 0.2866 (0.1662–0.4071) |
| $\quad IC_{50,2}$ | 0.533 (0.3205–0.7456) | 2.858 (1.731–3.985) | 1.265 (0.7121–1.819) | 0.04567 (0.02515–0.0662) |
| $R^2$ | 0.9523 | 0.9851 | 0.945 | 0.959 |

[a]$h_2$ and $IC_{50,2}$ represent the Hill coefficient and the $IC_{50}$ of camostat, brequinar, avoralstat, and nafamostat, respectively, for each drug combination.

against the delta VOC using a recently developed higher-order combination assay (23, 59). The triple combination conferred 100% antiviral efficacy at much lower concentrations of each drug in the cocktail than for any of the three pairwise combinations of the same constituent drugs (Fig. 6a). Moreover, the triple drug combination conferred a significantly higher synergy score than all three two-drug combinations (Fig. 6b). Compared to the two-drug combinations, the triple combination increased both the potency and efficacy of the antiviral effect yet showed no toxic effects on noninfected cells (see Fig. S6).

## DISCUSSION

Besides paxlovid and molnupiravir, there are presently no approved and easy to administer (e.g., oral, intranasal, inhaled) antiviral medications to prevent or ameliorate SARS-CoV-2 infection. Because SARS-CoV-2 continues to evolve rapidly (12–14), drug combination studies such as described in this report provide one step toward more effective drug-based control of the ongoing and possibly future pandemics. Effective drug combinations against HIV and HCV are comprised of two or three DAAs (18, 19). Similarly, we show that combining the two currently approved DAAs, molnupiravir plus paxlovid, confers synergistic suppression of SARS-CoV-2 infection in human lung cells, as we predicted (32), and as recently reported by others (60, 61), including a report that the combination of molnupiravir plus paxlovid appears superior to either drug alone in a mouse model of SARS-CoV-2 (62). We also show that combining HTAs with a DAA provides a similar level of antiviral synergy as an all-DAA combination: three TMPRSS2 inhibitors examined, when combined with molnupiravir, synergistically suppress WT SARS-CoV-2 and the VOC tested. Thus, TMPRSS2 is a potential target for an HTA-containing drug combination, as we have shown here with camostat, nafamostat, and avoralstat. Moreover, investigational inhibitors such as N-0385 (63) and enoxaparin (64) inhibit TMPRSS2 function by distinct mechanisms. In addition to TMPRSS2 cleavage, the SARS-CoV-2 Spike protein is also processed by the cellular enzyme furin. As such, novel furin inhibitors have been shown to synergistically inhibit SARS-CoV-2 infection when combined with camostat (65). Moreover, a phase 2 randomized, double-blind, placebo-controlled clinical trial of oral camostat showed accelerated overall symptom resolution despite no reduction in intranasal viral RNA (66), further supporting the notion that combining other drugs with camostat might enhance clinical and antiviral efficacy. Hence, new TMPRSS2 and furin inhibitors may be worth testing as drug combinations (67, 68).

**FIG 3** Legend (Continued)

molnupiravir (EIDD-1931) 2 h prior to infection with the SARS-CoV-2 beta (A) or delta (B) VOC at an MOI 0.01. (C) Nirmatrelvir and EIDD-1931 were added to Calu-3 cells 2 h before infection with the delta VOC at an MOI 0.01. After 96 h, the cell viability was measured by a CellTiter-Glo assay (Promega), and the antiviral efficacy was calculated as described in Materials and Methods. For each panel (A, B, and C), the left plot shows the percent inhibition of infection, while the right plot depicts a two-dimensional topograph that highlights the areas of synergy across the dose response matrix, including the MSA (light gray box).

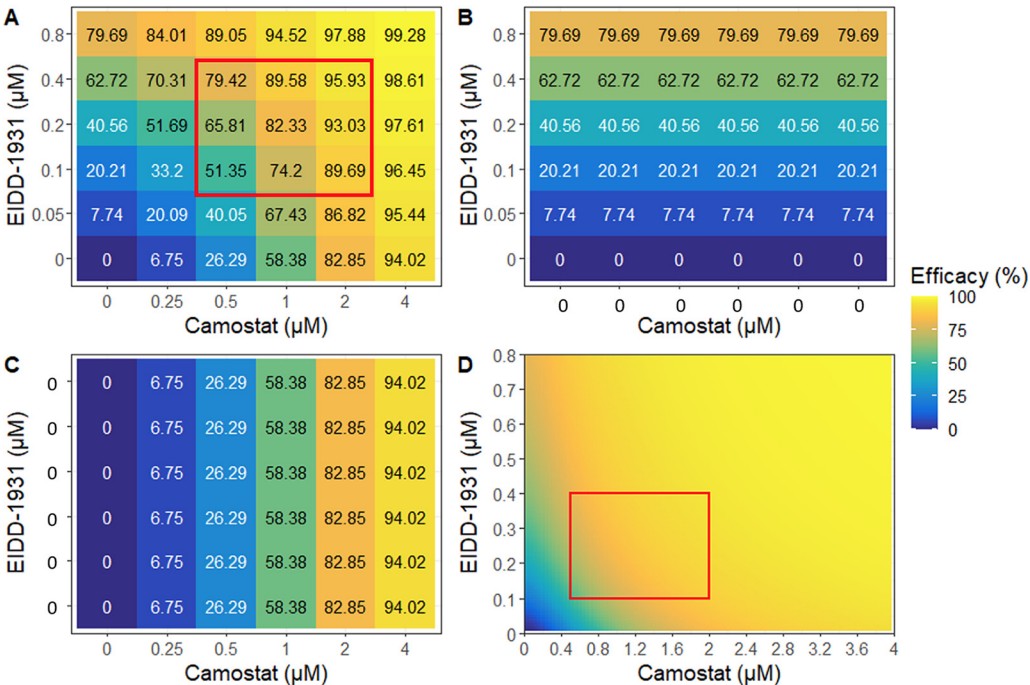

**FIG 4** PD modeling of molnupiravir plus camostat. (A) Model-projected efficacy of molnupiravir (EIDD-1931) plus camostat at empirically tested concentrations. (B) Projected efficacy of EIDD-1931 alone. (C) Projected efficacy of camostat alone. (D) Heat map of model projected inhibition at all combinatorial concentrations of both agents. The red box denotes the MSA.

Brequinar combined with molnupiravir conferred strong antiviral synergy against SARS-CoV-2 infection of Calu-3 cells. Our findings agree with recent studies showing that this HTA+DAA combination is synergistic *in vitro* and conferred superior antiviral effects in mice, although brequinar was administered via the intraperitoneal, not the oral, route in the recent report (31). Brequinar inhibits dihydroorotate dehydrogenase (DHODH), a ubiquitous host enzyme that is required for *de novo* pyrimidine synthesis (69). It was recently shown that brequinar, when combined with dipyridamole, an inhibitor of the pyrimidine salvage pathway (which can be activated when DHODH is inhibited by brequinar), also confers synergistic suppression of SARS-CoV-2 (70). Thus, dipyridamole may provide added benefit to brequinar-containing drug combinations.

We provide a mathematical model that faithfully reproduces experimental data and projects potency against SARS-CoV-2 at combinatorial drug concentrations that were not specifically tested experimentally. This model will serve as the basis for more comprehensive models that include PD data as shown here with human PK for the individual drugs and with SARS-CoV-2 dynamics, to accurately project human *in vivo* potency (71), as we recently did for synergistic drug combinations for Ebola virus (58). Combined PK, PD, and viral dynamics modeling can be conducted for the drug combinations tested here, as well as for others proposed in the literature (32, 72) or revealed in future studies. Lastly, with the triple combination of two HTAs (camostat and brequinar) and a DAA (molnupiravir), we demonstrate the potential of adding a third drug to enhance the potency of a drug pair, echoing the fact that certain drug regimens against HIV and HCV contain three drugs. The next stages of our work will involve further modeling (58) and testing of promising oral drug combinations in mouse models of SARS-CoV-2.

Paxlovid and molnupiravir have 89 and 30% efficacies, respectively, defined as prevention of hospitalization or death (7, 8, 73, 74). While current orally available drugs for SARS-CoV-2 are highly effective at reducing hospitalization, they are not effective for all indications (including postexposure prophylaxis). Although the issue of paxlovid

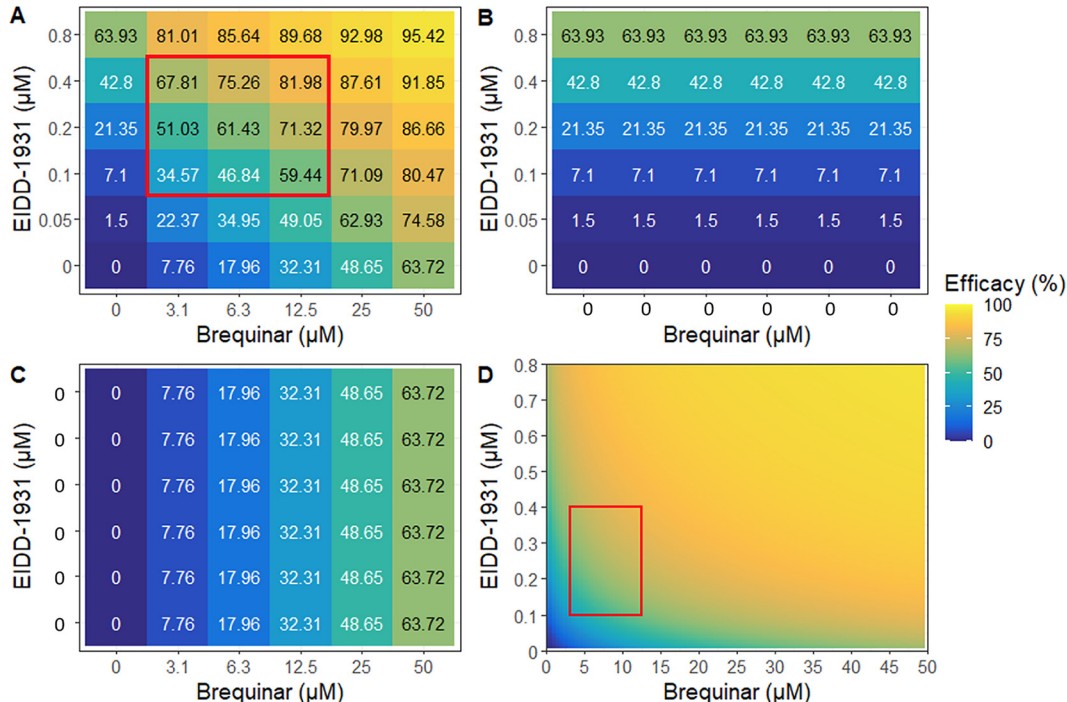

**FIG 5** PD modeling of molnupiravir plus brequinar. (A) Model projected efficacy of molnupiravir (EIDD-1931) plus brequinar at empirically tested concentrations. (B) Projected efficacy of EIDD-1931 alone. (C) Projected efficacy of brequinar alone. (D) Heat map of model projected inhibition at all combinatorial concentrations of both agents. The red box denotes the MSA.

resistance during monotherapy (12–16) has not yet fully emerged clinically, preexisting drug-resistant mutants have been found in therapy naive patients (75, 76). Moreover, resistance to remdesivir has also been observed *in vitro* (77, 78) and clinically (79). Thus, current antivirals are not guaranteed to maintain effectiveness against future variants. We propose that the drug combinations described here may demonstrate increased efficacy *in vivo* and may be of particular importance in immunocompromised

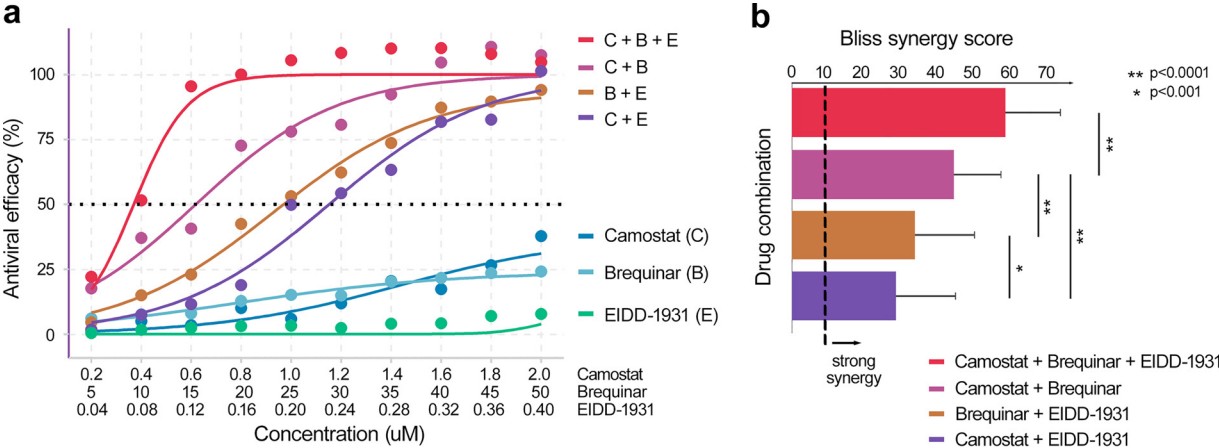

**FIG 6** Potent antiviral efficacy with a triple combination of host-targeting (HTA) and viral-targeting (DAA) drugs. (a) Camostat, brequinar, and molnupiravir (EIDD-1931) were mixed at top concentrations of 1.4, 50, and 0.28 $\mu$M, respectively, to produce one-, two-, or three-drug combinations. Stock concentrations were serially diluted in 10% increments and added to Calu-3 cells for 2 h prior to infection with the SARS-CoV-2 delta VOC at an MOI 0.01. After 96 h, the cell viability was measured by a CellTiter-Glo assay (Promega), and the antiviral efficacy was calculated as described in Materials and Methods. (b) Bliss synergy scores of the three-drug and two-drug combinations as calculated in SynergyFinder 3.0 (56). The data reflect the averages and standard deviations of triplicate samples per condition, from a single experiment, which was conducted twice with similar results. The dotted vertical line indicates the cutoff for strong Bliss synergy defined previously (23). *P* values are derived from two-sample *t* tests.

hosts who have a difficult time clearing the virus. As investigators are testing combinations of monoclonal antibodies plus a polymerase inhibitor for SARS-CoV-2 (80) and small molecule drug combinations for influenza (27), we advocate for further modeling and *in vivo* testing of oral small molecule drug combinations containing both HTAs and DAAs versus SARS-CoV-2, VOC, other coronaviruses, and other viruses for pandemic response and preparation (32).

## MATERIALS AND METHODS

**Chemicals, cell culture, and live virus.** Vero E6 and Calu-3 cells were maintained in standard medium (minimum essential medium; Gibco, catalog no. 11095) supplemented with 9% fetal bovine serum (FBS; HyClone, SH3007103) and 1% penicillin-streptomycin (Gibco, catalog no. 15140); HEK293T cells expressing human ACE2 and TMPRSS2 (55) (here called 293TAT cells), provided by Carol Weiss, were maintained in Dulbecco modified Eagle medium (DMEM; Gibco, catalog no. 11995), 9% FBS (HyClone, SH3007103), 1% penicillin-streptomycin (Gibco, catalog no. 15140), 1% nonessential amino acids (Gibco, catalog no. 11140), and 20 mM HEPES (Gibco, catalog no. 15630). HTA and DAA compounds were purchased from commercial vendors as outlined in Table S1 in the supplemental material.

Wild-type, infectious SARS-CoV-2 was obtained from BEI Resources (isolate USA-WA1/2020 NR-52281). The SARS-CoV_2 VOC stocks of beta and delta viruses were generated as follows. The beta virus (B.1.351 lineage), isolate hCoV-19/South Africa/KRISP-K005325/2020 (NR-54009; GISAID: EPI_ISL_678615) was originally obtained from BEI Resources, NIAID, NIH, and further expanded in Vero-TMPRSS2 cells in DMEM containing 2% heat-inactivated FBS (HI-FBS). The delta virus (GISAID accession number EPI_ISL_13858636) was isolated from local Seattle nasopharyngeal swab samples of a SARS-CoV-2-infected patient stored in viral transport medium, kindly provided by Alex Greninger. Briefly, the viruses in the VTM were first expanded using Vero-ACE2-TMPRSS2 cells in DMEM containing 2% HI-FBS. The initial crude expansion was then plaque-purified in Vero-ACE2-TMPRSS2 cells. The plaque-purified viruses were sent for amplicon sequencing (Swift Biosciences) to confirm their genome sequence. Virus was further amplified in Vero-TMPRSS2 cells. Virus titers were measured by plaque assays on Vero-TMPRSS2 cells.

**Drug treatment and infection of cells with live virus.** Compounds were added to cell plates and 2 h later, live virus (SARS-CoV-2 WA1 at a multiplicity of infection [MOI] of 0.1 or delta or beta VOC at MOIs of 0.01) was added to each infected well, or virus-free medium was added to each uninfected well. Plates were incubated at 37°C and 5% $CO_2$ for 48 or 96 h 293TAT cells or Calu-3 cells, respectively. The assay readout was the cell viability based on a CellTiter-Glo assay (Promega).

**Cell viability assay.** For drug dose-response studies, we measured cell viability in parallel wells using the CellTiter-Glo 2.0 assay (Promega). The assay measures the number of viable cells in culture by quantifying ATP, which indicates the presence of metabolically active cells. The contents of each well were aspirated, followed by the addition of 50 $\mu$L of phosphate-buffered saline (Gibco, catalog no. 10010031) and 50 $\mu$L of CellTiter-Glo reagent. The plate was then shaken for 2 min, incubated at room temperature for 10 min, and read for luminescence on a BioTek Synergy H4 plate reader.

**Drug combination assays.** Checkerboard assays of two drug combinations were performed as described earlier (23). Briefly, two drugs were tested in dose responses consisting of dimethyl sulfoxide (DMSO) control and five concentrations of each drug, yielding six concentrations of drug A and six concentrations of drug B. Combining the two separate drug dose responses creates a matrix or checkerboard of 36 dose-combinations of the two drugs. Calu-3 ($n = 50,000$) or 293TAT ($n = 10,000$) cells were seeded into each well of 96-well black, clear-bottom plates in the cell line's respective medium. Each two-drug concentration tested in a 6 × 6 checkerboard assay was performed in duplicate for 293TAT cells or triplicate for Calu-3 cells. Cells were infected, and parallel wells were not infected to assess the toxicity of the drugs. Both plates were incubated at 37°C and 5% $CO_2$ for 48 h (for 293TAT cells) or 96 h (for Calu-3 cells) prior to the CTG assay. For three drug combination studies, we employed a method that samples the diagonal space of the three-dimensional checkerboard (23, 59). Briefly, camostat, brequinar, and molnupiravir (EIDD-1931) were mixed at top concentrations of 1.4, 50, and 0.28 $\mu$M, respectively, to produce one-, two-, and three-drug combinations. Stock concentrations were serially diluted in 10% increments and added to Calu-3 cells at 2 h prior to infection. After 96 h, cell viability was measured by a CellTiter-Glo assay (Promega).

**Data analyses.** For single-drug experiments, drug concentrations were log transformed, and the concentrations of drug(s) that inhibited virus by 50% (i.e., $IC_{50}$) and the concentrations of drug(s) that killed 50% of cells (i.e., $CC_{50}$) were determined via nonlinear logistic regressions of log(inhibitor) versus response-variable dose-response functions (four parameters) constrained to zero bottom asymptote by statistical analysis using Prism 9 (GraphPad Software, Inc.), as described previously (23). Input data for these equations consisted of relative light units (RLU) generated by CellTiter-Glo assays. The selectivity index was calculated by dividing the $CC_{50}$ by the $IC_{50}$. The antiviral efficacy was calculated by comparing the RLU from virus-infected (IFX) cells treated with drug to the average of the infected cells treated with solvent (DMSO) and expressed as a percentage relative to the virus-induced cytopathic effect: RLU of noninfected/DMSO-treated cells – RLU in infected DMSO treated cells, using the following equation:

$$\left( \frac{(\text{IFX/drug} - \text{average IFX/DMSO})}{(\text{average non-IFX/DMSO} - \text{average IFX/DMSO})} \right) \times 100$$

Cell viability was calculated by comparing the RLU from noninfected cells treated with drugs to the noninfected cells treated with DMSO:

$$\left( \frac{\text{non-IFX/drug}}{\text{average non-IFX/DMSO}} \right) \times 100$$

Dose-response data from checkerboard assays were analyzed in SynergyFinder3, an open-access platform for multidrug combination synergies (56). Several combination parameters were reported from SynergyFinder3, including the average Bliss Synergy Score of the entire dose-response matrix and the maximum synergistic area (MSA), which corresponds to the maximum Bliss score calculated over an area of nine doses of the two compounds in a checkerboard experiment (i.e., $3 \times 3$ dose-response submatrix). Synergy is defined as when the observed inhibition is greater than that predicted by multiplicative Bliss independence at a given set of drug concentrations. For triple-drug experiments, the three-drug combination was compared to the three two-drug combinations of the drugs that comprise the triple combination using SynergyFinder3.0. Finally, SynToxProfiler (57) was used to evaluate the toxicity profile of the drug combinations in noninfected cells via checkerboard assays. For this, toxicity was calculated by subtracting the viability results above from 100 (to convert to percent inhibition values), and checkerboard toxicity data were uploaded to SynToxProfiler (see Fig. S2).

**Mathematical modeling.** We previously modified the Bliss independence PD model (81) to directly assess the potency of drug combinations across different drug concentrations. The purpose of the model was to recapitulate observed experimental data and then make projections about combinatorial drug potency at concentrations that were not measured experimentally. The model can then be synchronized with PK models to predict the percentage of new cell infections being prevented at any point during the drug dosing interval. The model includes the parameters $IC_{50}$ (the drug concentration at which 50% of infections are prevented), Hill coefficients of both drugs ($h_1$ and $h_2$, the slope of the dose-response curve), and an exponent ($a$) for the assessment of synergy (58), as expressed in the following equation:

$$\text{Efficacy}_{\text{combo}} = 100 \times \left( \frac{1}{1 + \left(\frac{D_1}{IC_{50,1}}\right)^{h1}} \times \frac{1}{1 + \left(\frac{D_2}{IC_{50,2}}\right)^{h2}} \right)^{a}$$

## SUPPLEMENTAL MATERIAL

Supplemental material is available online only.

**SUPPLEMENTAL FILE 1**, PDF file, 1.6 MB.

## ACKNOWLEDGMENTS

This article is dedicated to the memory of Hugo R. Rosen, a talented clinician-scientist, collaborator, colleague, and friend.

J.M.W. was partially supported by NIH grant AI114776. S.J.P. was partially supported by a Washington Research Foundation Technology Commercialization Grant and the Department of Laboratory Medicine and Pathology (University of Washington). T.A. was partially supported by Academy of Finland grants 345803 and 340141. S.B.B. was partially supported as an Open Philanthropy Awardee of the Life Sciences Research Foundation. M.G. and T.-Y.H. were supported by NIH/NIAID grant AI151698. S.P. was supported by BMBF grants (01KI2006D, 01KI20328A, and 01KX2021), by the Ministry for Science and Culture of Lower Saxony (14-76103-184, MWK HZI COVID-19), and by the German Research Foundation (DFG; PO 716/11-1 and PO 716/14-1).

We thank Carol Weiss for the 293TAT cells and Alex Greninger for the delta clinical sample. The following reagent was deposited by the Centers for Disease Control and Prevention and obtained through BEI Resources, NIAID, NIH: SARS-related coronavirus 2, isolate USA-WA1/2020, NR-52281. The following reagent was obtained through BEI Resources, NIAID, NIH: SARS-related coronavirus 2, isolate hCoV-19/South Africa/KRISP-K005325/2020, NR-54009, contributed by Alex Sigal and Tulio de Oliveira.

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
