## [Reviewer comments · Microbiology Spectrum]

Microbiology Spectrum

Combinations of host- and virus-targeting antiviral drugs confer synergistic suppression of SARS-CoV-2

Jessica Wagoner, Shawn Herring, Tien-Ying Hsiang, Aleksandr Ianevski, Scott Biering, Shuang Xu, Markus Hoffmann, Stefan Pöhlmann, Michael Gale, Jr., Tero Aittokallio, Joshua Schiffer, Judith White, and Stephen Polyak

Corresponding Author(s): Stephen Polyak, University of Washington

Review Timeline:

Submission Date:

September 6, 2022

Accepted:

September 12, 2022

Editor: Miguel Martinez

Reviewer(s): The reviewers have opted to remain anonymous.

Transaction Report:

DOI: <https://doi.org/10.1128/spectrum.03331-22>

September 12, 2022

Dr. Stephen J Polyak
University of Washington
Seattle

Re: Spectrum03331-22 (Combinations of host- and virus-targeting antiviral drugs confer synergistic suppression of SARS-CoV-2)

Dear Dr. Stephen J Polyak:

Your manuscript has been accepted, and I am forwarding it to the ASM Journals Department for publication. You will be notified when your proofs are ready to be viewed.

Sincerely,

Miguel Martinez
Editor, Microbiology Spectrum

Journals Department
Table S1, Table S2, Figs S1-S6: Accept